# A deep learning method for simultaneous denoising and missing wedge reconstruction in cryogenic electron tomography

Simon Wiedemann[1] & Reinhard Heckel [1] ✉

Cryogenic electron tomography is a technique for imaging biological samples in 3D. A microscope collects a series of 2D projections of the sample, and the goal is to reconstruct the 3D density of the sample called the tomogram. Reconstruction is difficult as the 2D projections are noisy and can not be recorded from all directions, resulting in a missing wedge of information. Tomograms conventionally reconstructed with filtered back-projection suffer from noise and strong artefacts due to the missing wedge. Here, we propose a deep-learning approach for simultaneous denoising and missing wedge reconstruction called DeepDeWedge. The algorithm requires no ground truth data and is based on fitting a neural network to the 2D projections using a self-supervised loss. DeepDeWedge is simpler than current state-of-the-art approaches for denoising and missing wedge reconstruction, performs competitively and produces more denoised tomograms with higher overall contrast.

Cryogenic electron tomography (cryo-ET) is a powerful cryo-electron microscopy (cryo-EM) technique for obtaining 3D models of biological samples such as cells, viruses, and proteins. An important application of cryo-ET is visualizing biological macromolecules like proteins in situ, i.e. in their (close-to) native environment. Imaging in situ preserves biological context, which can greatly improve the understanding of the workings of macromolecules[1].

In cryo-ET, the sample to be imaged is first prepared on a grid and then frozen. Next, a transmission electron microscope records a tilt series, which is a collection of 2D projections of the sample's 3D scattering potential or density. Each projection in the tilt series is recorded after tilting the sample for a number of degrees around the microscope's tilt axis.

From this tilt series, a tomogram, i.e. a discretized estimate of the sample's 3D density, can be estimated using computational techniques. For this inverse problem, numerous approaches have been proposed. The most commonly used tomographic reconstruction technique is filtered back-projection (FBP)[1–3]. Two major obstacles limit the resolution and interpretability of tomograms reconstructed with FBP and similar methods:

1. Noisy data: The total electron dose during tilt series acquisition must be low because biological samples are sensitive to radiation damage. Thus, the individual projections of the tilt series have low contrast and a low signal-to-noise ratio (SNR).

2. A missing wedge of information: The range of angles at which useful images can be collected is often limited to, for example, $\pm 60°$ rather than the desired full range of $\pm 90°$. This is due to the increased thickness of the sample in the direction of the electron beam for tilt angles of large magnitude[4]. The missing data in the tilt series is wedge-shaped in the Fourier domain.

To define the missing wedge and motivation for our work, we now state a model for the tilt series acquisition process in cryo-ET: a tilt series $\mathbf{t} = (\mathbf{t}_{-K}, \ldots, \mathbf{t}_K)$ is a collection of 2D projections, where each 2D projection $\mathbf{t}_k$ is a measurement of an underlying ground truth volume $\mathbf{v}^*$ obtained with the electron microscope. By the Fourier slice theorem (see e.g.[5]), the 2D Fourier transform $\mathbf{Ft}_k$ of each projection image $\mathbf{t}_k$ of a tilt series is a noisy observation of a 2D central slice through the true volume's 3D Fourier transform $\mathbf{Fv}^*$ multiplied with an additional filter, i.e.

$$\mathbf{Ft}_k = \mathbf{C}_k \cdot \mathbf{SR}_k \mathbf{Fv}^* + \mathbf{Fn}_k. \qquad (1)$$

[1]Department of Computer Engineering, Technical University of Munich, Munich, Germany. ✉e-mail: reinhard.heckel@tum.de

In Eq. (1), the rotation operator $\mathbf{R}_k$ spatially rotates the 3D Fourier transform $\mathbf{Fv}^*$ of the volume $\mathbf{v}^*$ by the tilt angle $\alpha_k$ around the microscope's tilt axis. Then, the slice-operator $\mathbf{S}$ extracts the central 2D slice of the volume's Fourier transform that is perpendicular to the microscope's optical axis. The filter $\mathbf{C}_k$ is the contrast transfer function (CTF) of the microscope and models optical aberrations. The term $\mathbf{Fn}_k$ is the Fourier transform of a random 2D image-domain noise term $\mathbf{n}_k$. It is often assumed that the noise $\mathbf{n}_k$ comes from a Poisson distribution (shot noise). Another common assumption is that the Fourier-domain noise $\mathbf{Fn}_k$ is Gaussian, but not necessarily white[6,7]. In this work, we make the weaker assumption that the 2D noise terms $\mathbf{n}_k$ and $\mathbf{n}_\ell$ of any two distinct projections indexed with $k$ and $\ell$ ($k \neq \ell$) have zero means and are independent.

Equation (1) is commonly used as a Fourier-domain model of the image formation process in cryo-EM[6,8]. As the Fourier slice theorem assumes continuous representations of volumes and images, this model is exact only in the continuous case. In this work, we consider discrete representations of volumes and images, as is common in cryo-EM practice. Thus, Eq. (1) holds only approximately.

The range of tilt angles $\alpha_k$ that yield useful projections $\mathbf{t}_k$ is typically limited to, e.g. ±60° rather than the full range ±90°. Therefore, by Eq. (1) there is a wedge-shaped region of the sample's Fourier representation $\mathbf{Fv}^*$ which is not covered by any of the Fourier slices $\mathbf{Ft}_k$.

We consider the problem of reconstructing a tomogram, i.e. a discretized estimate of a sample's density from a noisy, incomplete tilt series. This is a challenging inverse problem due to the high noise level and the missing wedge.

Recently, deep-learning-based methods for denoising and missing wedge reconstruction have been proposed. However, these are effective for denoising and missing wedge reconstruction individually but not simultaneously. Specifically, IsoNet[4], a closely related work, does well at missing wedge reconstruction, but its denoising performance is low compared to state-of-the-art denoising methods[9]. The current state-of-the-art approaches for denoising in cryo-ET build on Noise2Noise[10], a framework for deep-learning-based image denoising. Popular software packages that implement Noise2Noise-based denoising methods for cryo-ET tomograms are CryoCARE[11], Topaz[12] and Warp[13].

In this paper, we propose DeepDeWedge, a deep learning-based approach for tomogram reconstruction that simultaneously performs well on denoising and missing wedge reconstruction.

DeepDeWedge takes one (or more; see the "Results" section) tilt series as input and aims to estimate a noise-free 3D reconstruction of the samples' density with a filled-in missing wedge. To achieve this, we propose fitting a randomly initialized network with a self-supervised loss for simultaneous denoising and missing wedge reconstruction. After fitting, the network is applied to the same data to estimate the tomogram. DeepDeWedge only uses the tilt series of the density we wish to reconstruct and no other training data. DeepDeWedge is most related to Noise2Noise-based denoising approaches and to IsoNet, both introduced above. We discuss the exact relations in the "Results" section after describing our algorithm. DeepDeWedge performs on par with IsoNet on a pure missing wedge reconstruction problem and achieves state-of-the-art denoising performance on a pure denoising problem. Moreover, we find that the performance of DeepDeWedge for the joint denoising and wedge reconstruction problem is similar to the two-step approach of applying IsoNet to tomograms denoised with a state-of-the-art Noise2Noise-like denoiser. Moreover, DeepDeWedge is simpler and requires fewer hyperparameters to tune than the two-step approach.

## Results
In this section, we describe DeepDeWedge and evaluate its effectiveness on real and simulated data.

## The DeepDeWedge algorithm
DeepDeWedge takes a single tilt series $\mathbf{t}$ as input and produces a denoised, missing-wedge-filled tomogram. The method can also be applied to a dataset containing multiple (typically up to 10) tilt series from different samples of the same type, for example, sections of different cells, which share the same cell type. DeepDeWedge consists of the following three steps, illustrated in Fig. 1:

1. Step: Data preparation. First, split the tilt series $\mathbf{t}$ into two sub-tilt-series $\mathbf{t}^0$ and $\mathbf{t}^1$ with the even/odd split or the frame-based split. The even/odd split partitions the tilt series into even and odd projections based on their order of acquisition. The frame-based split can be applied if the tilt series is collected using dose fractionation and entails averaging only the even and odd frames recorded at each tilt angle. We recommend the frame-based splitting approach whenever possible. After splitting, we have two sub-tilt series.

   Next, reconstruct both sub-tilt-series independently with FBP and apply CTF correction. This yields a pair of two coarse reconstructions (FBP($\mathbf{t}^0$), FBP($\mathbf{t}^1$)) of the sample's 3D density. Finally, use a 3D-sliding window procedure to extract $N$ overlapping cubic sub-tomogram pairs $\{(\mathbf{v}_i^0, \mathbf{v}_i^1)\}_{i=1}^N$ from both FBP reconstructions. The size and number $N$ of these sub-tomogram cubes is a hyperparameter. Experiments on synthetic data presented in the supplementary information suggest that larger sub-tomograms tend to yield better results up to a point.

2. Step: Model fitting. Fit a randomly initialized network $f_\theta$, we use a U-Net[14], with weights $\boldsymbol{\theta}$ by repeating the following steps until convergence:

   (a) Generate model inputs and targets: For each of the sub-tomogram pairs $\{(\mathbf{v}_i^0, \mathbf{v}_i^1)\}_{i=1}^N$ generated in Step 1, sample a rotation $\mathbf{R}_{\varphi_i}$ parameterized by Euler angles $\varphi_i$ from the uniform distribution on the group of 3D rotations, and construct a model input $\tilde{\mathbf{v}}_{i,\varphi_i}^0$ and target $\mathbf{v}_{i,\varphi_i}^1$ by applying the rotation $\mathbf{R}_{\varphi_i}$ to both sub-tomograms and adding an artificial missing wedge to the rotated sub-tomogram $\mathbf{R}_{\varphi_i}\mathbf{v}_i^0$, as shown in the centre panel of Fig. 1. The missing wedge is added by taking the Fourier transform of the rotated sub-tomograms and multiplying them with a binary 3D mask $\mathbf{M}$ that zeros out all Fourier components that lie inside the missing wedge. Repeating this procedure for all sub-tomogram pairs $\{(\mathbf{v}_i^0, \mathbf{v}_i^1)\}_{i=1}^N$ yields a set of $N$ triplets consisting of model input, target sub-tomogram, and angle $\{(\tilde{\mathbf{v}}_{i,\varphi_i}^0, \mathbf{v}_{i,\varphi_i}^1, \varphi_i)\}_{i=1}^N$.

   (b) Update the model: Update the model weights $\boldsymbol{\theta}$ by performing gradient steps to minimize the per-sample loss

   $$\ell(\boldsymbol{\theta}, i) = \left\| \left( \mathbf{M}\mathbf{M}_{\varphi_i} + 2\mathbf{M}^C\mathbf{M}_{\varphi_i} \right) \mathbf{F} \left( f_\theta\left(\tilde{\mathbf{v}}_{i,\varphi_i}^0\right) - \mathbf{v}_{i,\varphi_i}^1 \right) \right\|_2^2. \quad (2)$$

   Here, $\mathbf{M}_{\varphi_i}$ is the rotated version of the wedge mask $\mathbf{M}$ and $\mathbf{M}^C := \mathbf{I} - \mathbf{M}$ is the complement of the mask. For the gradient updates, we use the Adam optimizer[15] and perform a single pass through the $N$ model input and target sub-tomograms generated before.

   (c) Update the missing wedges of the model inputs: For each, $i = 1, \ldots, N$, update the missing wedge in the $i$-th sub-tomogram $\mathbf{v}_i^0$ produced in Step 1 by passing it through the current model $f_\theta$, and inserting the predicted content of the missing wedge, as follows

   $$\mathbf{v}_i^0 \leftarrow \mathbf{F}^{-1}\left( \mathbf{M}\mathbf{F}\mathbf{v}_i^0 + \mathbf{M}^C\mathbf{F}f_\theta(\mathbf{v}_i^0) \right). \quad (3)$$

   In the next input-target generation step, the model inputs $\{\tilde{\mathbf{v}}_{i,\varphi_i}^0\}_{i=1}^N$ are constructed using the updated sub-tomograms. We do not update the missing wedges of the sub-tomograms $\{\mathbf{v}_i^1\}_{i=1}^N$ used to generate the model targets, since their missing

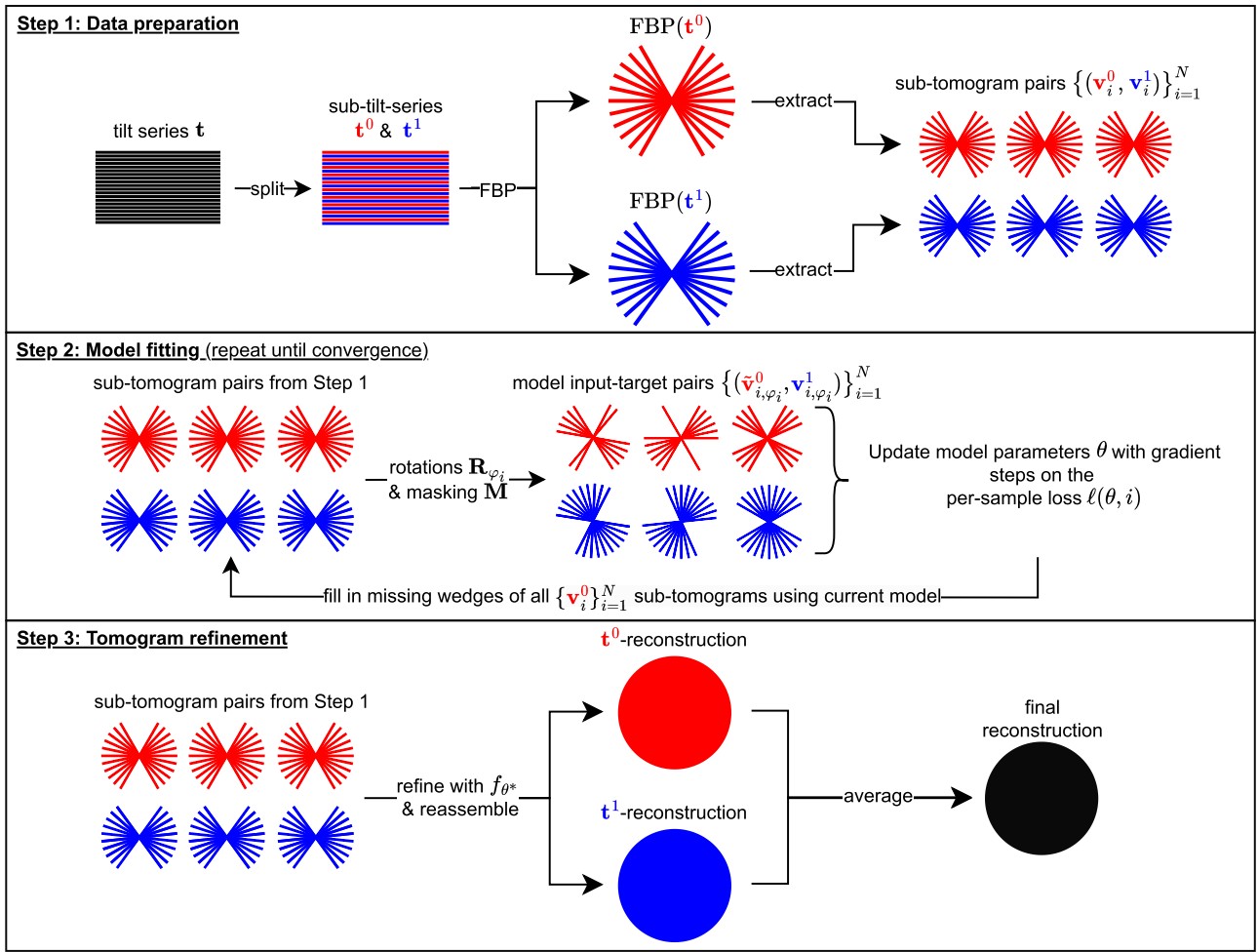

**Fig. 1 | Illustration of DeepDeWedge.** For simplicity, we show the 2D tilt series images as 1D Fourier slices and all 3D tomograms as 2D objects in the Fourier domain. Recall that tilt series images, tomograms, and sub-tomograms are objects in the image domain. The figure shows the splitting approach where the tilt series is split into even and odd projections.

wedges are masked out in the per-sample loss, c.f. Eq. (2), and therefore play no role in model fitting.

3. Step: Tomogram refinement. Pass the original, non-updated sub-tomograms $\{\mathbf{v}_i^0\}_{i=1}^N$ through the fitted model $f_{\boldsymbol{\theta}^*}$ from Step 2, and reassemble the model outputs $\{f_{\boldsymbol{\theta}^*}(\mathbf{v}_i^0)\}_{i=1}^N$ into a full-sized tomogram. Repeat the same for the sub-tomograms $\{\mathbf{v}_i^1\}_{i=1}^N$. Finally, average both reconstructions to obtain the final denoised and missing-wedge corrected tomogram.

## Motivation for the three steps of the algorithm

In Step 1, we split the tilt series into two disjoint parts to obtain measurements with independent noise. As the noise on the individual projections or frames is assumed to be independent, the reconstructions FBP ($\mathbf{t}^0$) and FBP ($\mathbf{t}^1$) are noisy observations of the same underlying sample with independent noise terms. Those are used in Step 2 for the self-supervised Noise2Noise-inspired loss.

Tilt series splitting is also used in popular implementations of Noise2Noise-like denoising methods for cryo-ET[11–13]. The frame-based splitting procedure was proposed by Buchholz et al.[11], who found that it can improve the performance of Noise2Noise-like denoising over the even/odd split.

Step 2 of DeepDeWedge is to fit a neural network to perform denoising and missing wedge reconstruction, for which we have

designed a specific loss function $\ell$ (Eq. (2)). We provide a brief justification for the loss function here; a detailed theoretical motivation is presented in the following section.

As the masks $\mathbf{M}\mathbf{M}_{\varphi_i}$, and $\mathbf{M}^C\mathbf{M}_{\varphi_i}$ are orthogonal, the loss value $\ell(i, \boldsymbol{\theta})$ can be expressed as the sum of two the two terms $\|\mathbf{M}\mathbf{M}_{\varphi_i}\mathbf{F}(f_\theta(\tilde{\mathbf{v}}_{i,\varphi_i}^0) - \mathbf{v}_{i,\varphi_i}^1)\|_2^2$, and $\|2\mathbf{M}^C\mathbf{M}_{\varphi_i}\mathbf{F}(f_\theta(\tilde{\mathbf{v}}_{i,\varphi_i}^0) - \mathbf{v}_{i,\varphi_i}^1)\|_2^2$. The first summand, i.e. $\|\mathbf{M}\mathbf{M}_{\varphi_i}\mathbf{F}(f_\theta(\tilde{\mathbf{v}}_{i,\varphi_i}^0) - \mathbf{v}_{i,\varphi_i}^1)\|_2^2$, is the squared L2 distance between the network output and the target sub-tomogram $\mathbf{v}_{i,\varphi_i}^1$ on all Fourier components that were not masked out by the two missing wedge masks $\mathbf{M}$ and $\mathbf{M}_{\varphi_i}$. As we assume the noise in the target to be independent of the noise in the input $\tilde{\mathbf{v}}_{i,\varphi_i}^0$, minimizing this part incentivizes the network to learn to denoise these Fourier components. This is inspired by the Noise2Noise principle.

The second summand, i.e. $\|2\mathbf{M}^C\mathbf{M}_{\varphi_i}\mathbf{F}(f_\theta(\tilde{\mathbf{v}}_{i,\varphi_i}^0) - \mathbf{v}_{i,\varphi_i}^1)\|_2^2$, incentivizes the network $f_\theta$ to restore the data that we artificially removed with the mask $\mathbf{M}$, and can be considered as a Noisier2Noise-like loss[16] (see Supplementary Information 1 for background). For this part, it is important that we rotate both volumes, which moves their original missing wedges to a new, random location.

In the last part of Step 2, we correct the missing wedges of the sub-tomograms $\{\mathbf{v}_i^0\}_{i=1}^N$ using the current model $f_\theta$. Therefore, as the model fitting proceeds, the model inputs $\{\tilde{\mathbf{v}}_{i,\varphi_i}^0\}_{i=1}^N$ will more and more

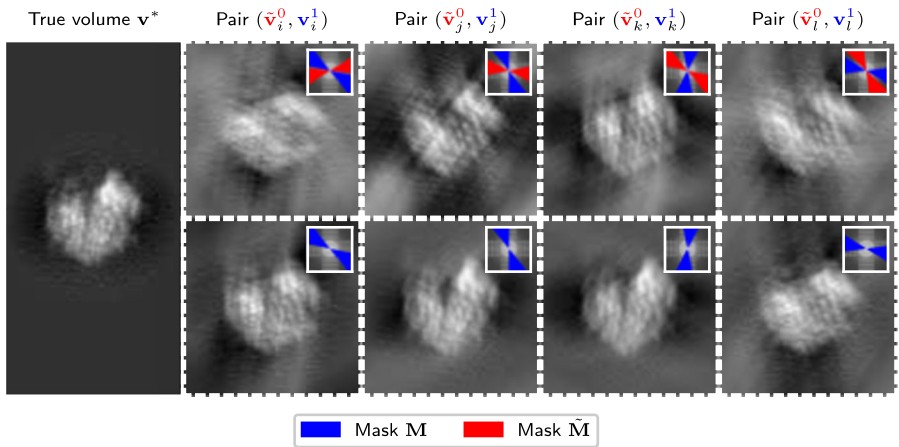

**Fig. 2 | Illustration of a single ground-truth structure v\*, and four model input-target pairs corresponding to four different random positions of the original (blue) and additional (red) missing wedge for the setup of our theoretical motivation.** For simplicity, we visualize a 2D structure with no additive noise and random in-plane rotations. The inset boxes in each patch show the absolute values of the Fourier transforms of the images and regions that are zeroed out by the missing wedge masks.

resemble sub-tomograms with only one missing wedge, i.e. the one we artificially remove from the partially corrected sub-tomograms $\{v_i^0\}_{i=1}^N$ with the mask **M**. This is a heuristic which is supposed to help the model perform well on the original sub-tomograms $\{v_i^0\}_{i=1}^N$ and $\{v_i^1\}_{i=1}^N$, which have only one missing wedge and which we use as model inputs in Step 3. An analogous approach for Noisier2Noise-based image denoising was proposed by Zhang et al.[17].

In Step 3, we use the fitted model from Step 2 to produce the final denoised and missing wedge-filled tomogram. To use all the information contained in the tilt series **t** for the final reconstruction, we separately refine the sub-tomograms from the FBP reconstructions $FBP(t^0)$ and $FBP(t^1)$ of the sub-tilt-series $t^0$ and $t^1$, and average them. For this, we apply a special normalization to the model inputs, which is described in Supplementary Information 2.

## Theoretical motivation for the loss function
Here, we present a theoretical result that motivates the choice of our per-sample loss $\ell$ defined in Eq. (2). The discussion in this section does not consider the heuristic of updating the missing wedges of the model inputs, which is part of DeepDeWedge's model fitting step. Moreover, we consider an idealized setup that deviates from practice to motivate our loss.

We assume access to data that consists of many noisy, missing-wedge-affected 3D observations of a fixed ground truth 3D structure $v^* \in \mathbb{R}^{N \times N \times N}$. Specifically, data is generated as two measurements (in the form of volumes) of the unknown ground-truth volume of interest, $v^*$, as

$$v^0 = \mathbf{F}^{-1}\mathbf{MF}(v^* + n^0), \qquad v^1 = \mathbf{F}^{-1}\mathbf{MF}(v^* + n^1), \qquad (4)$$

where $n^0, n^1 \in \mathbb{R}^{N \times N \times N}$ are random noise terms and $\mathbf{M} \in \{0,1\}^{N \times N \times N}$ is the missing wedge mask. From the first measurement, we generate a noisier observation $\tilde{v}^0 = \mathbf{F}^{-1}\tilde{\mathbf{M}}\mathbf{F}v^0$ by applying a second missing wedge mask $\tilde{\mathbf{M}}$. The noisier observation has two missing wedges: the wedge introduced by the first and the wedge introduced by the second mask. We assume that the two masks follow a joint and symmetric distribution, e.g. that for each mask, a random wedge is chosen uniformly at random. Figure 2 illustrates four data points in 2D.

We then train a neural network $f_\theta$ to minimize the loss

$$L(\boldsymbol{\theta}) = \mathbb{E}_{\mathbf{M},\tilde{\mathbf{M}},n^0,n^1}\left[\left\|\left(\tilde{\mathbf{M}}\mathbf{M} + 2\tilde{\mathbf{M}}^C\mathbf{M}\right)\mathbf{F}\left(f_\theta\left(\tilde{v}^0\right) - v^1\right)\right\|_2^2\right], \qquad (5)$$

where the expectation is over the random masks and the noise terms. Note that this resembles training on infinitely many data points, with a very similar loss to the original loss (2); the main difference is that in the original loss, the volume is rotated randomly but the mask **M** is fixed, while in the setup considered in this section, the volume is fixed but the masks **M** and $\tilde{\mathbf{M}}$ are random.

After training, we can use the network to estimate the ground-truth volume by applying the network to another noisy observation $\tilde{v}^0$. The following proposition, whose proof can be found in Supplementary Information 4, establishes that this is equivalent to training the network on a supervised loss to reconstruct the input $\tilde{v}^0$, provided the two masks are non-overlapping.

**Proposition 1.** Assume that the noise $n^1$ is zero-mean and independent of the noise $n^0$, and of the masks $(\mathbf{M}, \tilde{\mathbf{M}})$, and assume that the noise $n^0$ is also independent of the masks. Moreover, assume that the joint probability distribution $P$ of the missing wedge masks **M** and $\tilde{\mathbf{M}}$ is symmetric, i.e. $P(\mathbf{M}, \tilde{\mathbf{M}}) = P(\tilde{\mathbf{M}}, \mathbf{M})$, and that the missing wedges do not overlap. Then the loss L is proportional to the supervised loss

$$R(\boldsymbol{\theta}) = \mathbb{E}_{\mathbf{M},\tilde{\mathbf{M}},n^0}\left[\left\|f_\theta\left(\tilde{v}^0\right) - v^*\right\|_2^2\right], \qquad (6)$$

i.e. $L(\boldsymbol{\theta}) = R(\boldsymbol{\theta}) + c$, where $c$ is a numerical constant independent of the network parameters $\boldsymbol{\theta}$.

In practice, we do not apply our approach to the problem of reconstructing a single fixed structure $v^*$ from multiple pairs of noisy observations with random missing wedges. Instead, we consider the problem of reconstructing several unique biological samples using a small dataset of tilt series. To this end, we fit a model with an empirical estimate of risk similar to the one considered in Proposition 1. We fit the model on sub-tomogram pairs extracted from the FBP reconstructions of the even and odd sub-tilt series, which exhibit independent noise. Moreover, as already mentioned above, in the setup of our algorithm, the two missing wedge masks **M** and $\tilde{\mathbf{M}}$ themselves are not random. However, as we randomly rotate the model input sub-tomograms during model fitting, the missing wedges appear at a random location with respect to an arbitrary fixed orientation of the sub-tomogram.

## Related work
DeepDeWedge builds on Noise2Noise-based denoising methods and is related to the denoising and missing wedge-filling method IsoNet[4]. We

first discuss the relation between DeepDeWedge and Noise2Noise-based methods, which do not reconstruct the missing wedge.

Noise2Noise-based denoising algorithms for cryo-ET as implemented in CryoCARE[11] or Warp[13] take one or more tilt series as input and return denoised tomograms. A randomly initialized network $f_\theta$ is fitted for denoising on sub-tomograms of FBP reconstructions $FBP(\mathbf{t}^0)$ and $FBP(\mathbf{t}^1)$ of sub-tilt-series $\mathbf{t}^0$ and $\mathbf{t}^1$ obtained from a full tilt series $\mathbf{t}$. The model is fitted by minimizing a loss function (typically the mean-squared error), between the output of the model $f_\theta$ applied to one noisy sub-tomogram and the corresponding other noisy sub-tomogram. The fitted model is then used to denoise the two reconstructions $FBP(\mathbf{t}^0)$ and $FBP(\mathbf{t}^1)$, which are then averaged to obtain the final denoised tomogram. Contrary to those denoising methods, DeepDeWedge fits a network not only to denoise, but also to fill the missing wedge.

Our method is most closely related to IsoNet, which can also do denoising and missing wedge reconstruction. IsoNet takes a small set of, say, one to ten tomograms and produces denoised, missing-wedge corrected versions of those tomograms. Similar to DeepDeWedge, a randomly initialized network is fitted for denoising and missing wedge reconstruction on sub-tomograms of these tomograms, however, the fitting process is different: Inspired by Noisier2Noise, the model is fitted on the task of mapping sub-tomograms that are further corrupted with an additional missing wedge and additional noise onto their non-corrupted versions. After each iteration, the intermediate model is used to predict the content of the original missing wedges of all sub-tomograms. The predicted missing wedge content is inserted into all sub-tomograms, which serve as input to the next iteration of the algorithm.

Different to IsoNet, our denoising approach is Noise2Noise-like, as in CryoCARE. This leads to better denoising performance, as we will see later, as well as requiring fewer assumptions and no hyperparameter tuning. Specifically, Noisier2Noise-like denoising requires knowledge of the noise model and strength (see Supplementary Information 1). As this knowledge is typically unavailable, Liu et al.[4] propose approximate noise models from which the user has to choose.

After model fitting, the user must manually decide which iteration and noise level gave the best reconstruction. Thus, IsoNet's Noisier2Noise-inspired denoising approach requires several hyperparameters for which good values exist but are unknown. Therefore, IsoNet requires tuning to achieve good results. Our denoising approach introduces no additional hyperparameters and does not require knowledge of the noise model and strength.

The main commonality between DeepDeWedge and IsoNet is the Noisier2Noise-like mechanism for missing wedge reconstruction, which consists of artificially removing another wedge from the sub-tomograms and fitting the model to reconstruct the wedge.

Moreover, like IsoNet, DeepDeWedge fills in the missing wedges of the model inputs. In IsoNet, one also has to fill in the missing wedges of the model targets. This is necessary because, contrary to our loss $\ell$ form Eq. (2), IsoNet's loss function does not ignore the targets' missing wedges via masking in the Fourier domain.

Another line of work related to DeepDeWedge considers domain-specific tomographic reconstruction methods that incorporate prior knowledge of biological samples into the reconstruction process to compensate for missing wedge artefacts, for example, ICON[18], and MBIR[19]. For an overview of such reconstruction methods, we refer to the introductory sections of works by Ding et al.[20] and Bohning et al.[21]. Liu et al.[4] found that IsoNet outperforms both ICON and MBIR.

DeepDeWedge is also conceptually related to un-trained neural networks, which reconstruct an image or volume based on fitting a neural network to data[22,23]. Un-trained networks also only rely on fitting a neural network to given measurements. However, they rely on the bias of convolutional neural networks towards natural images[24,25],

whereas in our setup, we fit a network on measurement data to be able to reconstruct from the same measurements.

For cryo-EM-related problems other than tomographic reconstruction, deep learning approaches for missing data reconstruction and denoising have also recently been proposed. Zhang et al.[26] proposed a method to restore the state of individual particles inside tomograms, and Liu et al.[27] proposed a variant of IsoNet to resolve the preferred orientation problem in single-particle cryo-EM[27].

## Experiments on purified *Saccharomyces cerevisiae* 80S ribosomes

In this and the following experiments, we compare DeepDeWedge to (a re-implementation of) CryoCARE, IsoNet, and a two-step approach of fitting IsoNet to tomograms denoised with CryoCARE. The two-step approach, which we call CryoCARE + IsoNet, is considered a state-of-the-art pipeline for denoising and missing wedge reconstruction.

The first dataset we consider is the commonly used EMPIAR-10045 dataset, which contains seven tilt series collected from samples of purified *S. cerevisiae* 80S Ribosomes.

Figure 3 shows a tomogram refined with IsoNet, CryoCARE + IsoNet and DeepDeWedge using the even/odd tilt series splitting approach. Note that while IsoNet's built-in Noisier2Noise-like denoiser removes some of the noise contained in the FBP reconstruction, its performance is considerably worse than that of Noise2Noise-based CryoCARE. This can be seen by comparing to the result of applying IsoNet with a disabled denoiser to a tomogram denoised with CryoCARE. DeepDeWedge produces a denoised and missing-wedge-corrected tomogram similar to the CryoCARE + IsoNet combination. The main difference between these reconstructions is that the DeepDeWedge-refined tomogram has a smoother background and contains fewer high-frequency components.

Regarding missing wedge correction, we find the performance of DeepDeWedge and IsoNet to be similar. In slices parallel to the $x$–$z$-plane, where the effects of the missing wedge on the FBP reconstruction are most prominent, both IsoNet and DeepDeWedge reduce artefacts and correct artificial elongations of the ribosomes. The central $x$–$z$-slices through the reconstructions' Fourier transforms confirm that all methods but FBP fill in most of the missing wedge.

## Experiments on flagella of *Chlamydomonas reinhardtii*

Next, we evaluate DeepDeWedge on another real-world dataset of tomograms of the flagella of *C. reinhardtii*, which is the tutorial dataset for CryoCARE. Since we observed above that IsoNet performs better when applying it to CryoCARE-denoised tomograms, we compare only to CryoCARE + IsoNet. In addition, we investigate the impact of splitting the tilt-series into even and odd projections versus using the frame-based split for DeepDeWedge and CryoCARE + IsoNet.

The reconstructions obtained with all methods are shown in Fig. 4. We find that when using the even/odd-based split, CryoCARE + IsoNet produces a crisper reconstruction than DeepDeWedge (see zoomed-in region). This may be because the model inputs and targets of IsoNet stem from denoised FBP reconstruction of the full tilt series in which the information is more densely sampled than in the sub-tomograms of the even and odd FBP to the reconstructions used for fitting DeepDeWedge.

When using the frame-based splitting method, in which DeepDeWedge also operates on the more densely sampled FBP reconstruction, DeepDeWedge removes more noise than CryoCARE + IsoNet and produces higher contrast, which is most noticeable in the $x$–$z$-slice. Especially in background areas, the DeepDeWedge reconstruction has fewer high-frequency components and is smoother. Therefore, the CryoCARE + IsoNet reconstruction is slightly more faithful to the FBP reconstruction but is also noisier.

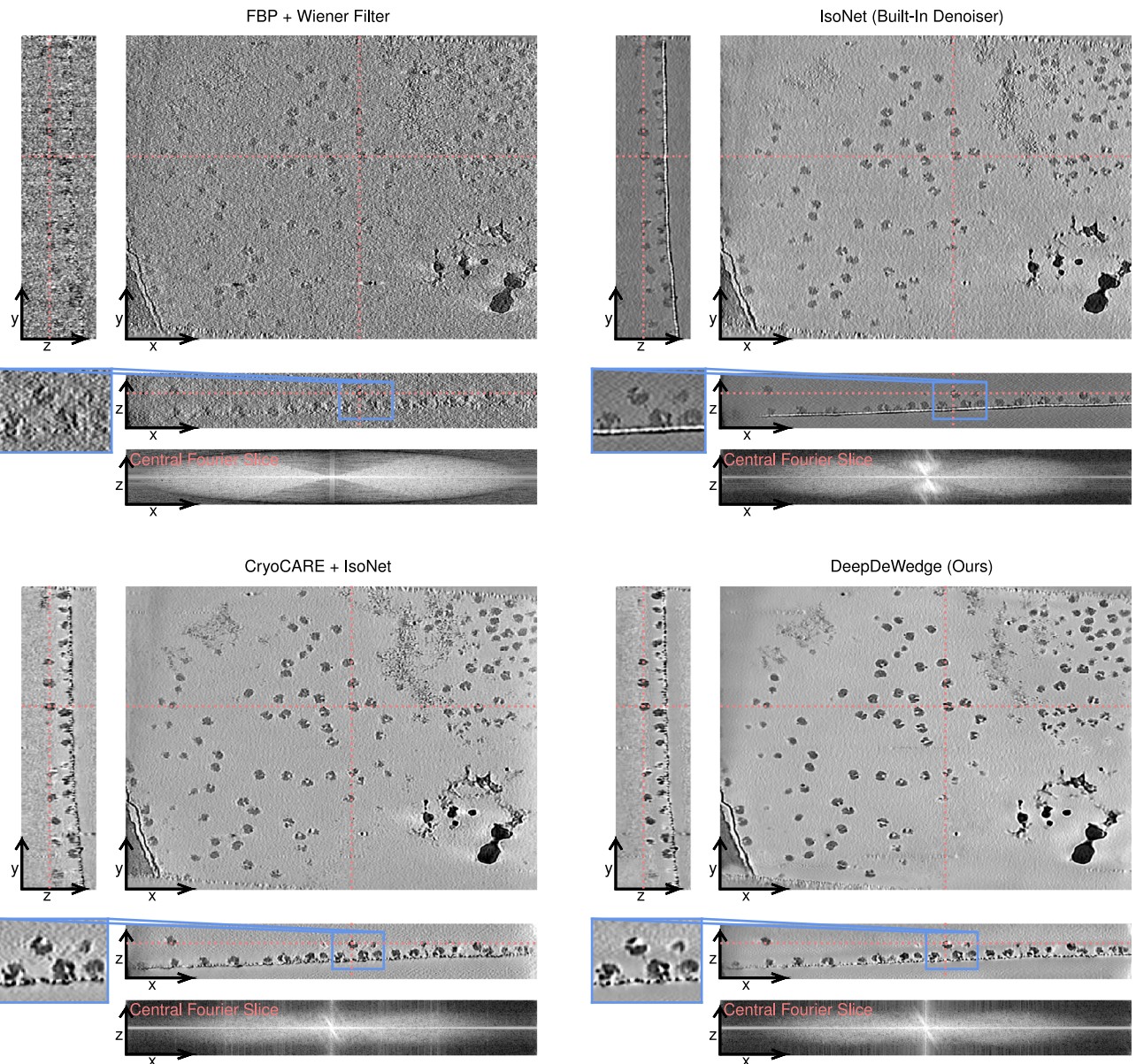

**Fig. 3 | Slices through 3D reconstructions of a tomogram containing purified *S. cerevisiae* 80S ribosomes (EMPIAR-10045, tomogram 5).** The red lines in each slice indicate the positions of the remaining two slices. We also show the central x–z-slice through the logarithm of the magnitude of the Fourier transform of each reconstruction.

The central x–z-slices through the reconstructions' Fourier transforms' indicate that both CryoCARE + IsoNet and DeepDeWedge fill in most of the missing wedge. Both methods fix the missing-wedge-caused distortions of the microtubules exhibited by the FBP reconstruction, as seen in the x–z-slices. DeepDeWedge reconstructs more of the flagellas' outer parts.

### Experiments on the ciliary transit zone of *C. reinhardtii*

Finally, we apply DeepDeWedge to an in situ dataset. We chose EMPIAR-11078[28], which contains a tilt series of the ciliary transit zone of *C. reinhardtii*. The crowded cellular environment and low contrast and SNR of the tilt series make EMPIAR-11078 significantly more challenging for denoising and missing wedge reconstruction than the two datasets from our previous experiments.

Slices through reconstructions of two tomograms obtained with FBP, CryoCARE + IsoNet and DeepDeWedge are shown in Fig. 5. In the x–z- and z–y-planes, the DeepDeWedge reconstructions are more crisp and less noisy than those produced with CryoCARE + IsoNet. Especially

in the x–z-plane, where the effects of the missing wedge are strongest, DeepDeWedge produces higher contrast than CryoCARE + IsoNet and removes more of the artefacts. Again, the CryoCARE + IsoNet reconstructions contain more high-frequency components and are closer to the FBP reconstruction, whereas the DeepDeWedge reconstructions are smoother and more denoised, especially in empty or background regions.

Remarkably, as can be seen in the zoomed-in regions in the second row of Fig. 5, both CryoCARE + IsoNet and DeepDeWedge reconstruct parts of the sample that are barely present in the FBP reconstruction since they are perpendicular to the electron beam direction, which means that a large portion of their Fourier components are masked out by the missing wedge.

Note that both CryoCARE + IsoNet and DeepDeWedge appear less effective at reconstructing the missing wedge compared to the two experiments presented above. This is indicated by the central x–z-slices through the Fourier transform of the reconstructions and is likely due to the challenging, crowded nature and low SNR of the data.

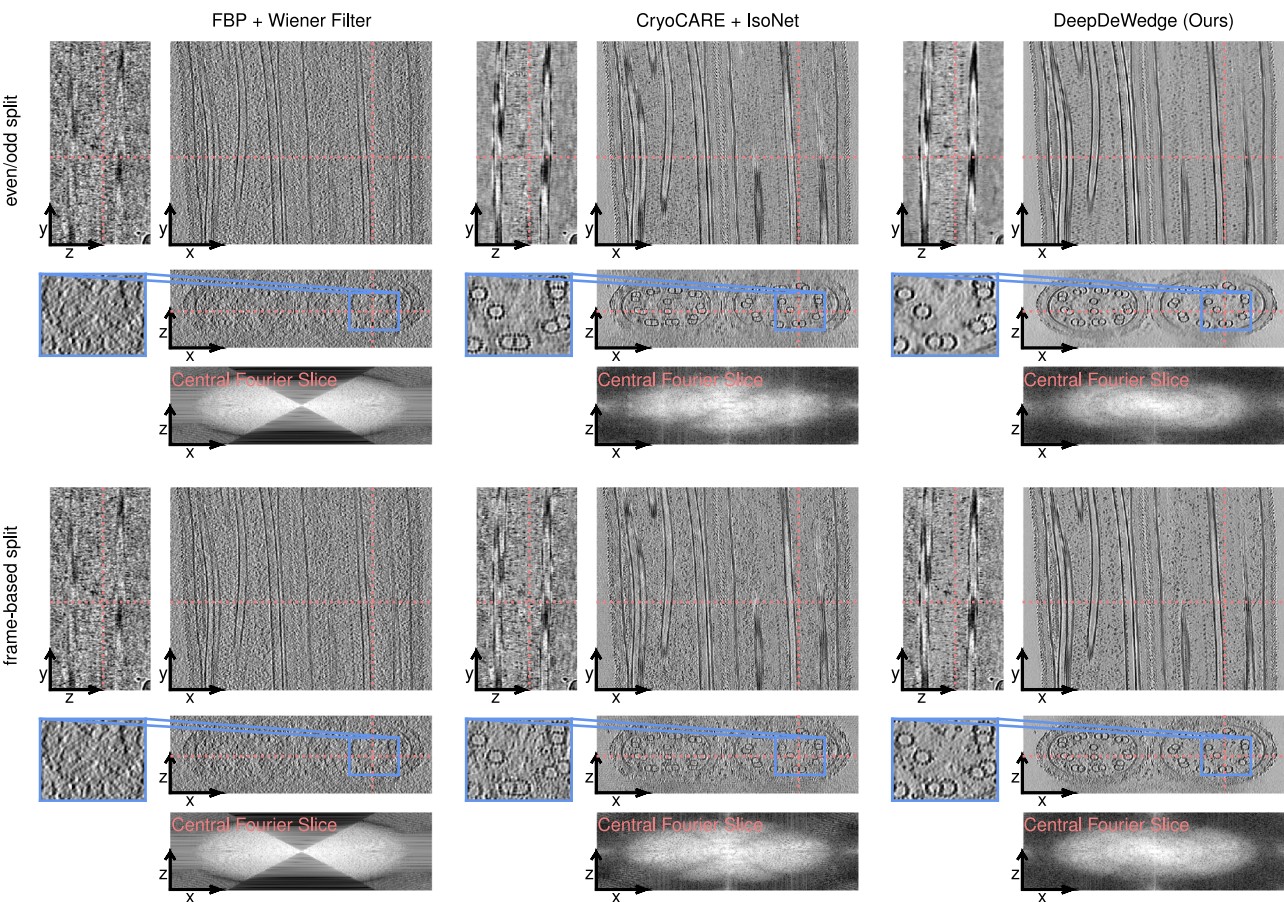

**Fig. 4 | Slices through 3D reconstructions of the flagella of *C. reinhardtii* when using different reconstruction methods.** The red lines in each slice indicate the positions of the remaining two slices. We also show the central *x*–*z*-slice through the logarithm of the magnitude of the Fourier transform of each reconstruction.

## Experiments on synthetic data

Here, we compare DeepDeWedge to IsoNet, CryoCARE, and the CryoCARE + IsoNet combination on synthetic data to quantify our qualitative findings on real data.

We used a dataset by Gubins et al.[29] containing 10 noiseless synthetic ground truth volumes with a spatial resolution of 10 Å$^3$/voxel and a size of 179 × 512 × 512 voxels. All volumes contain typical objects found in cryo-ET samples, such as proteins (up to 1500 uniformly rotated samples from a set of 13 structures), membranes, and gold fiducial markers.

For our comparison, we fitted models on the first three tomograms of the SHREC 2021 dataset. We used the Python library tomosipo[30] to compute clean projections of size 512 × 512 in the angular range ±60° with 2° increment. From these clean tilt series, we generated datasets with different noise levels by adding pixel-wise independent Gaussian noise to the projections. We simulated three datasets with tilt series SNR 1/2, 1/4, and 1/6, respectively.

To measure the overall quality of a tomogram $\hat{v}$ obtained with any of the three methods, we calculated the normalized correlation coefficient $CC(\hat{v}, v^*)$ between the reconstruction $\hat{v}$ and the corresponding ground truth $v^*$, which is defined as

$$CC(\hat{v}, v^*) = \frac{\langle \hat{v} - \text{mean}(\hat{v}), v^* - \text{mean}(v^*) \rangle}{\|\hat{v} - \text{mean}(\hat{v})\|_2 \|v^* - \text{mean}(v^*)\|_2}. \quad (7)$$

By definition, it holds that $0 \le CC(\hat{v}, v^*) \le 1$, and the higher the correlation between reconstruction and ground truth, the better. The correlation coefficient measures the reconstruction quality (both the denoising and the missing wedge reconstruction capabilities) of the

methods. To isolate the denoising performance of a method from its ability to reconstruct the missing wedge, we also report the correlation coefficient between the refined reconstructions and the ground truth after applying a 60° missing wedge filter to both of them. We refer to this metric as "CC outside the missing wedge" and it is used to compare the denoising performance of CryoCARE, which does not perform missing wedge reconstruction.

As a central application of cryo-ET is the analysis of biomolecules, we also report the resolution of all proteins in the refined tomograms. For this, we extracted all proteins from the ground truth and refined tomograms and calculated the average 0.143 Fourier shell correlation cutoff (0.143-FSC) between the refined proteins and the ground truth ones. The 0.143-FSC is commonly used in cryo-EM applications. Its unit is Angstroms, and it seeks to express up to which spatial frequency the information in the reconstruction is reliable. In contrast to the correlation coefficient, a lower 0.143-FSC value is better. To measure how well each method filled in the missing wedges of the structures, we also report the average 0.143-FSC calculated only on the true and predicted missing wedge data. We refer to this value as (average) "0.143-FSC inside the missing wedge".

Figure 6 shows the metrics for decreasing SNR of the tilt series. All metrics suggest that DeepDeWedge yields higher-quality reconstructions than IsoNet, CryoCARE, and CryoCARE + IsoNet.

CryoCARE achieves a lower correlation coefficient than IsoNet in the high-SNR regime, while the order is reversed for low SNR. A likely explanation is that for lower noise levels, the correlation coefficient is more sensitive to the missing wedge artefacts in the reconstructions. CryoCARE does not perform missing wedge reconstruction, so it has a lower correlation coefficient than IsoNet for higher SNR. For lower

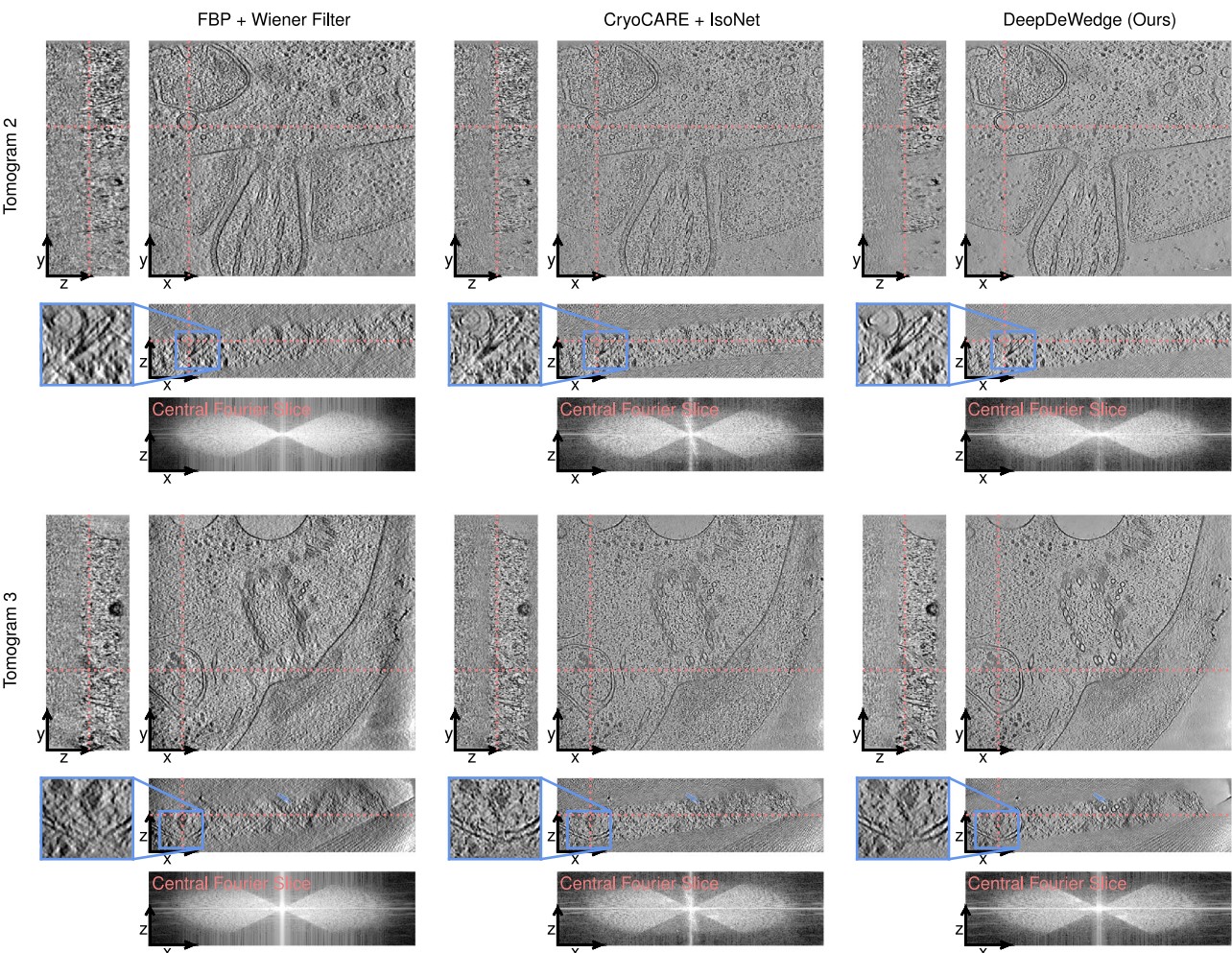

**Fig. 5 | Slices through 3D reconstructions of tomograms showing the ciliary transit zone of *C. reinhardtii*.** The red lines in each slice indicate the positions of the remaining two slices. We also show the central *x–z*-slice through the logarithm of the magnitude of the Fourier transform of each reconstruction. The row labels follow the naming convention of EMPIAR-11078.

SNR, the correlation coefficient is dominated by the noise. Regarding denoising, we and others[9] have observed that CryoCARE performs better than IsoNet, which is confirmed here by looking at the correlation coefficient outside the missing wedge. As expected, Cryo-CARE + IsoNet combines the strengths of both methods. Compared to this combination, the overall quality of DeepDeWedge reconstructions is on par or better, depending on the noise level.

The FSC metrics in the second row of Fig. 6 indicate that the average resolution of the proteins in the refined tomograms is approximately the same for IsoNet and DeepDeWedge and that they perform similarly for missing wedge reconstruction.

## Discussion

In this paper, we have proposed DeepDeWedge, which is a deep-learning-based method for denoising and missing wedge reconstruction in cryo-ET.

Compared to the state-of-the-art CryoCARE + IsoNet pipeline, DeepDeWedge removes more of the noise and, therefore, produces smoother reconstructions. While this occasionally results in the loss of some high-frequency details, DeepDeWedge produces overall cleaner reconstructions with higher contrast, especially of medium or large objects such as microtubules (Figs. 4 and 5), cellular structures (Figs. 5 and 7), and large proteins (Figs. 3 and 5). On the other hand, the CryoCARE + IsoNet results are more faithful to the FBP reconstructions, especially for high-frequency details, as can be seen in Fig. 5.

Regarding missing wedge reconstruction, which can be mainly seen in the removal of missing wedge artefacts and the reduction of artificial elongations, we find that DeepDeWedge and CryoCARE + IsoNet have overall similar performance. Both methods perform best for small and medium-sized objects like proteins or microtubules, whereas large structures like vesicles (Figs. 5 and 7) or the boundaries of the *C. reinhardtii* flagella (Fig. 4) pose greater challenges. This could be a result of using sub-tomograms for model fitting, as sub-tomograms may not always contain all the information that is necessary for missing wedge reconstruction of objects larger than the sub-tomogram size.

The data that DeepDeWedge fills in for the missing wedge is based on prior information about the data that the network learns during model fitting by predicting data that we artificially removed. However, it is important to keep in mind that the actual information contained in the missing wedge is irreversibly lost during tilt series acquisition.

Therefore, we advise users of our method to be cautious, especially regarding reconstructing objects that are mostly perpendicular to the electron beam. Many of the Fourier components that correspond to such objects are contained in the missing wedge region. Examples of objects that are almost completely masked by the missing wedge include thin membranes or elongated proteins perpendicular to the electron beam; we discuss a concrete example and how Deep-DeWedge handles it in Supplementary Information 6. It remains an open question to what extent missing-wedge-filled tomograms, whether based on deep learning or classical methods, can be trusted

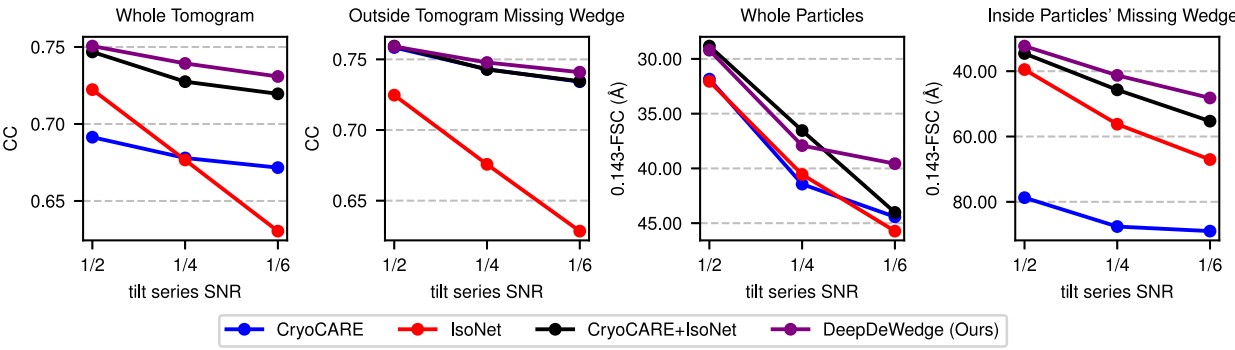

**Fig. 6 | Comparison of DeepDeWedge (with even/odd split) to IsoNet, CryoCARE and the combination of IsoNet and CryoCARE for increasing noise levels on the tilt series images.** We inverted the *y*-axes such that higher is better for all metrics.

(see also Supplementary Information 7). To estimate the trustworthiness of a DeepDeWedge reconstruction, we recommend comparing it to a reconstruction obtained with a classical, prior-free method that is highly data-consistent, such as FBP.

Based on these considerations, we see two main applications for DeepDeWedge: First, DeepDeWedge is an effective method to improve the direct interpretability of tomograms, which may enable discoveries that are prevented by high noise levels and strong missing wedge artefacts in raw FBP reconstructions. Second, tomograms reconstructed with DeepDeWedge can be used as input for downstream tasks such as segmentation or particle picking and might improve the effectiveness of those downstream tasks. It has already been observed that denoised and/or missing-wedge-corrected tomograms can improve the performance of deep-learning-based particle pickers[4,11,31]. For subsequent steps, such as sub-tomogram averaging, one can then use the original raw tomograms that are not modified by the neural network.

## Methods

Here, we describe the technical details of all the experiments presented in the previous section. First, we describe details that are common to all experiments, and then we discuss dataset-specific details.

We implemented DeepDeWedge in Python, using PyTorch[32] as a deep learning framework. For IsoNet, we used the authors' original implementation (https://github.com/IsoNet-cryoET/IsoNet), and for CryoCARE, we used our own re-implementation, which we based on our implementation of DeepDeWedge.

As model architecture, we used a U-Net[14] with 64 channels and three downsampling layers for all three methods. The network has 27.3 million trainable parameters and is also the default model architecture of IsoNet.

IsoNet uses regularization with dropout with a probability of 0.3, which we also kept for CryoCARE to prevent overfitting. For DeepDeWedge, we found that dropout is unnecessary, so we did not use dropout for our method unless explicitly stated otherwise.

We fitted all models using the Adam optimizer[15] with a constant learning rate of $4 \cdot 10^{-4}$.

### Experiments on purified *S. cerevisiae* 80S ribosomes

The EMPIAR-10045 dataset contains seven tilt series collected from samples of purified *S. cerevisiae* 80S Ribosomes. All tilt series are aligned and consist of 41 projections collected at tilt angles from −60° to +60° with a 3° increment.

- FBP and sub-tomograms: We performed all FBP reconstructions with a ramp filter in Python using the library tomosipo[30]. After

reconstruction, we downsampled all tomograms by a factor of six using average pooling, which resulted in a final voxel size of 13.02 Å. To these tomograms, we applied IsoNet's CTF deconvolution routine with parameters as described by Liu et al.[4] in their paper. For model fitting, we extracted sub-tomograms of shape 80 × 80 × 80. As wide regions of the tomograms contain only ice and no ribosomes, we used IsoNet's mask generation tool with the default parameters to produce a mask of the non-empty regions. After extracting the sub-tomograms, we selected only those that contained at least 40% sample according to the mask. For the refinement with the fitted model, we also used sub-tomograms of shape 80 × 80 × 80 voxels but extracted them without masking and with an overlap of 40 voxels.

- Details on IsoNet: We fitted IsoNet for 45 iterations (ten epochs per iteration) on 423 sub-tomograms. We used the default noise schedule for fitting, which starts adding noise with a level of 0.05 in iteration 11 and then increases the noise level by 0.05 every five iterations. We set the `noise_mode` parameter, which determines the distribution of the additional noise used for Noisier2Noise-like denoising, to `noFilter`, which corresponds to Gaussian noise. We also tried setting the noise mode to the `ramp`, as the tomograms were reconstructed with FBP with a ramp filter, but this gave worse results. In their paper, Liu et al.[4] fitted IsoNet for 30 iterations using similarly many sub-tomograms and the same number of epochs. However, we found that 45 iterations gave visually more appealing results compared to the images shown in the IsoNet paper.

- Details on CryoCARE + IsoNet: We first fitted CryoCARE on 1291 sub-tomograms for 200 epochs, which was approximately when the validation loss started to increase. Next, we fitted IsoNet for 50 iterations (ten epochs per iteration) on 418 sub-tomograms, and monitored the validation loss. Iteration 32 achieved the lowest validation loss, so we chose this model to produce the final reconstructions.

- Details on DeepDeWedge: We fitted the model on 418 sub-tomograms until the loss on a hold-out validation set did not improve anymore, which took about 1500 epochs. However, we also obtained reconstructions that were visually similar to the one shown in Fig. 3 around the 500 epoch mark.

### Experiments on flagella of *C. reinhardtii*

The dataset is a single tilt series collected from the flagella of *C. reinhardtii*. The projections were collected at angles from −65° to +65° with 2° increments. Each projection was acquired using dose fractionation with 10 frames per tilt angle. Each frame has a pixel size of 2.36 Å.

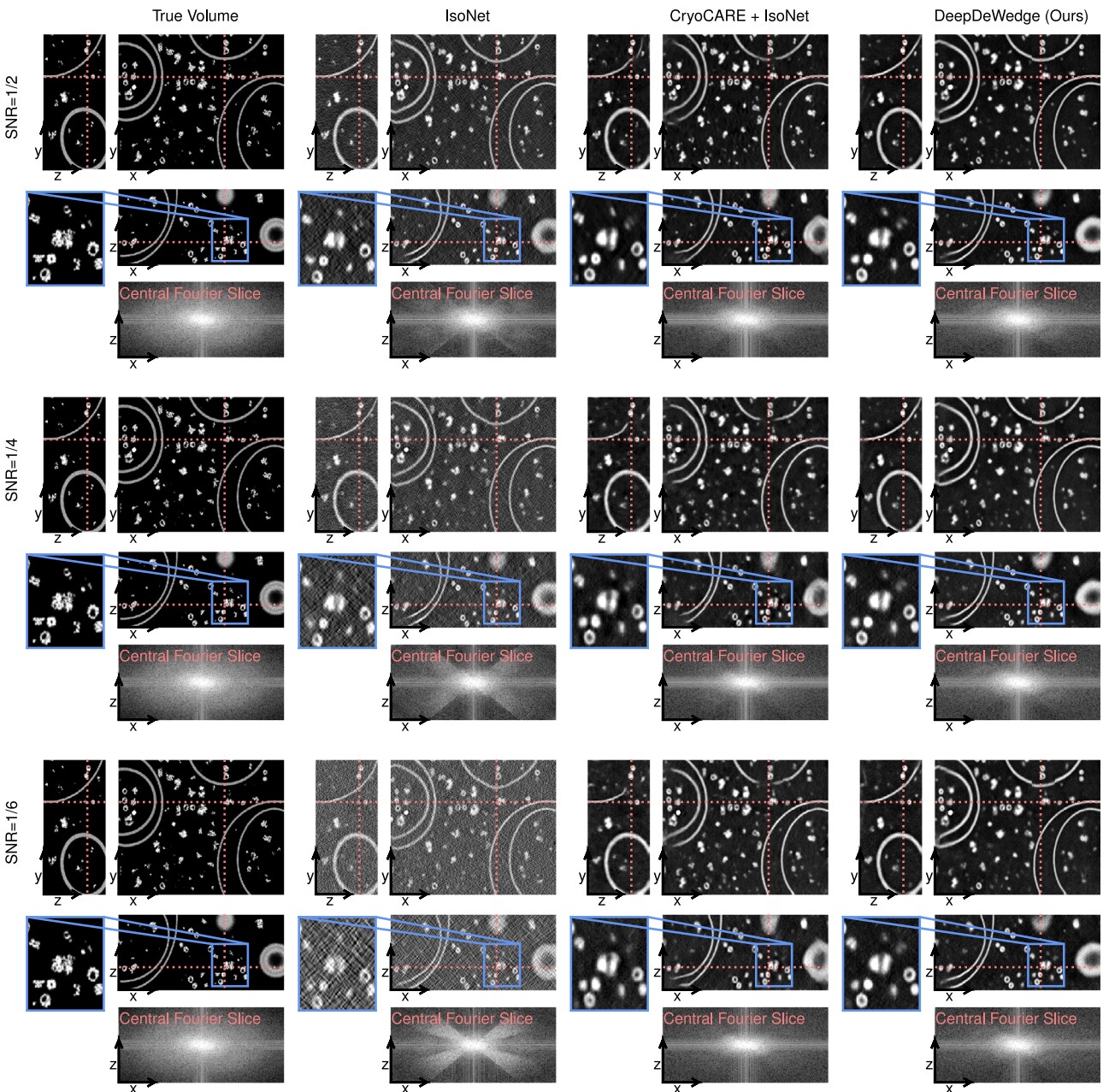

**Fig. 7 | Slices through tomograms reconstructed with CryoCARE + IsoNet and DeepDeWedge (with even/odd split) on a tomogram from the synthetic SHREC 2021 dataset.** The red lines in each slice indicate the positions of the remaining two slices. We also show the central $x$–$z$-slice through the logarithm of the magnitude of the Fourier transform of each reconstruction.

- FBP and sub-tomograms: We followed the steps described in the CryoCARE GitHub repository (https://github.com/juglab/cryoCARE_T2T/tree/master/example) to obtain a CTF-corrected tomogram from the tilt series with IMOD[33]. We downsampled the tomogram by a factor of 6 using average pooling, which resulted in a voxel size of 14.13 Å. As IsoNet is optimized for a missing wedge of 60°, we artificially widened the missing wedge from the original 50° to 60° by multiplying the reconstruction with a missing wedge filter in the Fourier domain for a fair comparison. We again extracted sub-tomograms with the shape $96 \times 96 \times 96$ for model fitting. For the final refinement, we used the same sub-tomogram size extracted with an overlap of 32 voxels.
- Details on CryoCARE + IsoNet: We first fitted CryoCARE for 2000 epochs on 252 sub-tomograms using early stopping on the self-supervised denoising loss on a hold-out validation set for model

selection. Next, we fitted IsoNet for 200 iterations (ten epochs per iteration) on 156 denoised sub-tomograms Interestingly, for the even/odd tilt series split, we found that the CryoCARE + IsoNet reconstruction looked better when using a CryoCARE model after 200 epochs rather than the one found with early stopping. In the main paper, we show these visually more appealing results.
- Details on DeepDeWedge: We fitted the model for 2000 epochs on 144 sub-tomograms.

### Experiments on the ciliary transit zone of *C. reinhardtii*
For our comparison, we chose tomograms 2, 3, and 8, since their tilt series were approximately collected at the angular range from −60° to +60° (increment: 2°), for which IsoNet was optimized. We fitted DeepDeWedge and the CryoCARE denoiser for CryoCARE + IsoNet using the frame-based split of the tilt-series data.

- FBP and sub-tomograms: We used IMOD to reconstruct the tomograms from the raw movie frames. First, we binned the pre-aligned tilt series 6 times, which resulted in a pixel size of 20.52 Å. Next, we performed phase flipping based on the provided defocus values for CTF correction and reconstructed the tomograms via FBP. We extracted sub-tomograms with the shape $96 \times 96 \times 96$ for model fitting. Again, we used masks generated with IsoNet to exclude empty areas from sub-tomogram extraction; each sub-tomogram had to contain at least 30% sample according to the mask. For the final refinement, we again used the same sub-tomograms of size $96 \times 96 \times 96$ extracted with an overlap of 32 voxels.
- Details on CryoCARE + IsoNet: We first fitted CryoCARE on 210 sub-tomograms for 2000 epochs, again using early stopping on the validation loss for model selection. Next, we fitted IsoNet for 300 iterations (ten epochs per iteration) on 264 denoised sub-tomograms.
- Details on DeepDeWedge: We fitted the model for 3000 epochs on 194 sub-tomograms.

## Experiments on synthetic data

In an effort to make the comparison between the methods as fair as possible, we tried to eliminate implementation details wherever possible. For example, we used our own custom implementation for subroutines like sub-tomogram extraction and reassembling sub-tomograms into full-sized volumes for all methods. Moreover, to be consistent with IsoNet's default parameters, we used dropout during model fitting in DeepDeWedge, although we found that it slightly deteriorated its performance.

- FBP and sub-tomograms: We performed FBP using tomosipo[30], this time with a Hamming-like filter (see below). We fitted all models for all methods using 150 sub-tomograms of size $96 \times 96 \times 96$. To refine the FBP reconstructions after model fitting, we applied the final fitted networks to sub-tomograms of the same size, using an overlap of 32 voxels.
- Details on CryoCARE: Despite the use of dropout, we observed overfitting, so we applied early stopping on a subset of the ground truth data to get an impression of the best-case performance. Depending on the noise level, the optimal early stopping lies between epochs 300 and 900.
- Details on IsoNet: As our simulated dataset does not contain CTF effects, we did not perform the CTF deconvolution preprocessing step, which is usually performed to increase the contrast of the FBP reconstructions and improve IsoNet's performance. We compensated for this by using the Hamming-like filter for FBP, which improves the contrast compared to the standard ramp filter. We fitted IsoNet for 50 iterations with 20 epochs per iteration. We again followed the default noise schedule described above and set the noise mode to `hamming`. As there is no principled way to determine when to stop IsoNet fitting or how much noise to add, we did the following to obtain a very strong baseline: We evaluated the IsoNet reconstructions after every iteration using all comparison metrics and chose the best result for each metric, which yields best-case performance. We emphasize that this approach is not possible in practice where ground truth is not available. The optimal performance of each model occurred for every SNR between iterations 30 and 40, i.e. between 600 and 800 epochs.
- Details on CryoCARE + IsoNet: For each noise level, we denoised all volumes used for model fitting with the early-stopped CryoCARE models we fitted before. Next, we fitted IsoNet for 50 iterations with 10 epochs per iteration and without the built-in Noisier2Noise-like denoiser. Finally, we again calculated all metrics for each iteration and reported the best ones.

- Details on DeepDeWedge: We fitted the network for 1000 epochs. At this point, the validation metrics were still improving, but only slightly.

## Reporting summary

Further information on research design is available in the Nature Portfolio Reporting Summary linked to this article.

## Data availability

All tomograms shown in Figs. 3–5 and 7 have been deposited to Figshare and are available at https://figshare.com/articles/journal_contribution/Tomograms_shown_in_Figures_of_the_DeepDeWedge_Paper/26169538. The EMPIAR-10045 dataset used in this study is available in the EMPIAR database under accession code 10045. The tilt series collected from the *C. reinhardtii* flagella is available at https://download.fht.org/jug/cryoCARE/Tomo110.zip. The EMPIAR-11078 dataset used in this study is available in the EMPIAR database under accession code 11078. The SHREC 2021 dataset is available at https://dataverse.nl/dataset.xhtml?persistentId=doi:10.34894/XRTJMA.

## Code availability

We provide an implementation and a tutorial of DeepDeWedge on GitHub: https://github.com/MLI-lab/DeepDeWedge.

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

## Acknowledgements
The authors would like to thank Tobit Klug, Kang Lin, Youssef Mansour, Ricardo Righetto, and Dave Van Veen for their helpful discussions. The authors would like to thank Ricardo Righetto for help regarding tomogram reconstruction from the movie frames in EMPIAR-11078. The authors are supported by the Institute of Advanced Studies at the Technical University of Munich, the Deutsche Forschungsgemeinschaft (DFG, German Research Foundation)—456465471, 464123524, the DAAD, the German Federal Ministry of Education and Research, and the Bavarian State Ministry for Science and the Arts. The authors also acknowledge the financial support by the Federal Ministry of Education and Research of Germany in the programme of "Souveraen. Digital. Vernetzt.". Joint project 6G-life, project identification number: 16KISK002.

## Author contributions
R.H. initiated and supervised the project. S.W. performed the experiments, developed the code, performed data analysis, and prepared illustrations. S.W. and R.H. wrote the manuscript.

## Funding

## Competing interests
The authors declare no competing interests.
