## [Peer Review File · Nature Communications]

A Deep Learning Method for Simultaneous Denoising and Missing Wedge Reconstruction in Cryogenic Electron TomographyREVIEWER COMMENTS

Reviewer #1 (Remarks to the Author):

Wiedemann and Heckel developed an algorithm that improves visual quality of the cryo-tomograms. Their method performs on par with IsoNet on a pure missing wedge reconstruction problem and achieves state-of-the-art denoising performance on a pure denoising problem. The remarkable contrast of tomograms achieved through DeepDeWedge is impressive. Nevertheless, I have two concerns related to the performance or reliability of the methods.

Major concerns:

1. The DeepDeWedge algorithm integrates the novel aspects of both IsoNet and cryoCARE. And it is generally admitted in the field that tomograms first processed with cryoCARE and then IsoNet will greatly enhance the interpretability of tomograms (<https://twitter.com/bengeliscious/status/1586725842264248320>). Because cryoCARE-IsoNet procedure are gradually becoming a routine in cryoET, it poses a question that whether and how DeepDeWedge can achieve a quality better than current cryoCARE-IsoNet practice, not IsoNet alone.
2. Lines 286-287, The author claimed DeepDeWedge showed details that are barely visible in IsoNet reconstruction. I observed that the highlighted details with blue arrows correspond to the air-water interface. Although some protein debris could adhere to this interface, the strong and thin-layered density in the DeepDeWedge-corrected tomogram should not exist in real data at all. It is possible that DeepDeWedge learned the features the 2-nm thick carbon layer and placed on the air-water interface because both interfaces can absorb ribosomes. The authors are likely to interpret artifacts generated by deep learning as real structures. This raise a concern that whether these results, though visually appealing, are not trustworthy in biology.

Minor concerns:

1. Line 16, I understand the algorithm requires no ground truth data, but why the algorithm does not require training data.
2. Lines 68-72, It is unclear how this algorithm is related to un-trained neural network. In this situation of missing wedge restoration, the network is not fitted to represent a particular image, and the bias are not coming from convolutional neural network itself.

3. Equation (2), based on what is written here, the mask M and $F(\cdot)$ are in Fourier space. I wonder how this mean squared error can be calculated as Fourier components. Could you explain whether complex-valued distances are calculated.

4. Line 377 “missing edge” should be “missing wedge”

5. Lines 391-392: The primary purpose of 'enhancing direct interpretability' is not solely to 'facilitate decisions on further analysis.' Its significance lies in directly facilitating scientific discoveries related to pleomorphic structures, such as membranes and organelles, within the complex cellular environment.

Reviewer #2 (Remarks to the Author):

The authors of the paper demonstrate a new method for cryoET tomogram noise reduction and elimination of the effects of missing wedge. This method integrates the algorithm ideas of the noise2noise model and the isonet method, and shows that it has better noise reduction performance than similar software. However, the methods proposed in the article needs to be further explained and tested in order to make sure the general applicability.

concerns :

1) In using a model similar to noise2noise, the algorithm needs to divide the data into odd and even data sets to calculate the tomogram separately. The signal-to-noise ratio of the cryoET data is already very low, and further splitting the data into two sets is bound to significantly reduce the signal-to-noise ratio of each set. Therefore, will the algorithm get a similar improvement in the processing of data with a lower signal-to-noise ratio, such as the data cut by cryoFIB, compared with isonet?

2) The testing of different types of cryoET data in this article is not comprehensive enough, and the data containing membrane structures, such as a cellular lamella, needs to be tested. Membrane structure, especially those perpendicular to the optical axis of the EM, is often challenging for missing wedge correction.

3) When the algorithm uses a method similar to isonet, it introduces additional or expanded missing wedges, which will inevitably adversely affect the convergence of the algorithm. Will this affect the stability of the algorithm?

minor concerns:

line 10: "the 2D projections have a missing wedge". Projections don't have missing wedge, only tomogram has missing wedge.

Responses to Reviewers' Comments

Reviewer comments are in **black**, **our** responses are in **blue**

We thank the reviewers very much for their valuable comments, based on which we substantially revised our paper and proposed method.

Reviewer #1 (Remarks to the Author):

Wiedemann and Heckel developed an algorithm that improves visual quality of the cryo-tomograms. Their method performs on par with IsoNet on a pure missing wedge reconstruction problem and achieves state-of-the-art denoising performance on a pure denoising problem. The remarkable contrast of tomograms achieved through DeepDeWedge is impressive. Nevertheless, I have two concerns related to the performance or reliability of the methods.

Major concerns:

1. The DeepDeWedge algorithm integrates the novel aspects of both IsoNet and cryoCARE. And it is generally admitted in the field that tomograms first processed with cryoCARE and then IsoNet will greatly enhance the interpretability of tomograms (<https://twitter.com/bengelicious/status/1586725842264248320>). Because cryoCARE-IsoNet procedure are gradually becoming a routine in cryoET, it poses a question that whether and how DeepDeWedge can achieve a quality better than current cryoCARE-IsoNet practice, not IsoNet alone.

We agree with the reviewer that a comparison between DeepDeWedge and the two-step CryoCARE+IsoNet approach is important. Therefore, we included results for CryoCARE+IsoNet in all our experiments for comparison to our method. As can be seen in the revised version of our paper, DeepDeWedge denoises at least as well as CryoCARE+IsoNet. Regarding missing wedge reconstruction, we found DeepDeWedge to perform better than the two-step approach in most cases.

2. Lines 286-287, The author claimed DeepDeWedge showed details that are barely visible in IsoNet reconstruction. I observed that the highlighted details with blue arrows correspond to the air-water interface. Although some protein debris could adhere to this interface, the strong and thin-layered density in the DeepDeWedge-corrected tomogram should not exist in real data at all. It is possible that DeepDeWedge learned the features the 2-nm thick carbon layer and placed on the air-water interface because both interfaces can absorb ribosomes. The authors are likely to interpret artifacts generated by deep learning as real structures. This raise a concern that whether these results, though visually appealing, are not trustworthy in biology.

We thank the reviewer for making us aware that our original DeepDeWedge version over-interpreted or even hallucinated features for this case. Based on this important observation and some additional experiments that confirmed that this was an over-interpreted or hallucinated feature, we slightly re-designed our method. As can be seen in the revised version of our paper (Figure 3 in Section 2.2.1, Figure E.1 in Appendix E), refining the ribosome tomogram with the new version of DeepDeWedge does not produce the unexpectedly strong density pointed out by the reviewer.

These artifacts in the first version originated from the mismatch of data used for model fitting, i.e., sub-tomograms from the even FBP reconstruction with an additional missing wedge, and the data used in the refinement step, where we used sub-tomograms from the full FBP reconstruction without

additional missing wedges. We designed the new version of DeepDeWedge, which reduces this mismatch as much as possible. Details on this can be found in Section 2.1, and Appendix G, where we comment on the hallucinations in the previous version.

Minor concerns:

1. Line 16, I understand the algorithm requires no ground truth data, but why the algorithm does not require training data.

Throughout the paper, we avoid the term “training” since DeepDeWedge is not a trained deep learning method in the usual sense, where a model is trained on a dedicated training set consisting of and then applied to arbitrary test data.

2. Lines 68-72, It is unclear how this algorithm is related to un-trained neural network. In this situation of missing wedge restoration, the network is not fitted to represent a particular image, and the bias are not coming from convolutional neural network itself.

The commonality of the two methods is that both un-trained networks and DeepDeWedge are used to reconstruct specific objects of interest by fitting a network to the measurement data without relying on external training data.

3. Equation (2), based on what is written here, the mask M and $F(\cdot)$ are in Fourier space. I wonder how this mean squared error can be calculated as Fourier components. Could you explain whether complex-valued distances are calculated.

When calculating the loss in Equation (2) in the Fourier domain, the 2-norm in Equation (2) is the complex 2-norm, which is supported in the deep learning framework PyTorch, in which DeepDeWedge is implemented.

However, in our implementation, we evaluated the loss in real space by taking the inverse Fourier transform of the term inside the norm. As this term is conjugate symmetric, the inverse Fourier transform yields a real-valued object (up to numerical precision). As the inverse Fourier transform preserves the 2-norm, both ways to calculate the loss give the same value.

4. Line 377 “missing edge” should be “missing wedge”

Thanks, we corrected this typo.

5. Lines 391-392: The primary purpose of 'enhancing direct interpretability' is not solely to 'facilitate decisions on further analysis.' Its significance lies in directly facilitating scientific discoveries related to pleomorphic structures, such as membranes and organelles, within the complex cellular environment.

Thanks for pointing this out. We added this to the list of primary applications of DeepDeWedge, as outlined in Section 3.

Reviewer #2 (Remarks to the Author):

The authors of the paper demonstrate a new method for cryoET tomogram noise reduction and elimination of the effects of missing wedge. This method integrates the algorithm ideas of the noise2noise model and the isonet method, and shows that it has better noise reduction performance than similar software. However, the methods proposed in the article needs to be further explained and tested in order to make sure the general applicability.

concerns:

1) In using a model similar to noise2noise, the algorithm needs to divide the data into odd and even data sets to calculate the tomogram separately. The signal-to-noise ratio of the cryoET data is already very low, and further splitting the data into two sets is bound to significantly reduce the signal-to-noise ratio of each set. Therefore, will the algorithm get a similar improvement in the processing of data with a lower signal-to-noise ratio, such as the data cut by cryoFIB, compared with isonet?

Section 2.2.3 of the revised version of our paper contains a comparison between CryoCARE+IsoNet and DeepDeWedge on a challenging in-situ dataset collected from the ciliary transit zone of *C. reinhardtii*, which was acquired from frozen cells thinned with FIB.

We found that DeepDeWedge removes more of the missing wedge artifacts than IsoNet and, at the same time, denoises slightly better than CryoCARE.

2) The testing of different types of cryoET data in this article is not comprehensive enough, and the data containing membrane structures, such as a cellular lamella, needs to be tested.

We hope that the additional experiment on cellular tomograms described in our answer to the first concern also helps to alleviate the second concern.

Membrane structure, especially those perpendicular to the optical axis of the EM, is often challenging for missing wedge correction.

We agree that reconstructing membranes and any other objects that are perpendicular to the microscope's optical axis is particularly difficult (and sometimes maybe even impossible), as most of the Fourier components that correspond to such objects are contained in the missing wedge. Nevertheless, we found that our method partially restores such objects, e.g., in the flagella (Section 2.2.2, Figure 4) and ciliary transit zone (Section 2.2.3, Figure 5, zoomed-in region in Tomogram 3) of *C. reinhardtii*.

3) When the algorithm uses a method similar to isonet, it introduces additional or expanded missing wedges, which will inevitably adversely affect the convergence of the algorithm. Will this affect the stability of the algorithm?

Below, we included a fitting (left) and validation (right) loss curve for applying DeepDeWedge on the ribosome tomograms (EMPIAR-10045). Both curves converge and indicate no significant instability in the fitting process.

minor concerns:

line 10: “the 2D projections have a missing wedge”. Projections don’t have missing wedge, only tomogram has missing wedge.

Thanks for pointing that out; we changed the statement to “[...] the 2D projections [...] cannot be recorded from all necessary viewing directions, resulting in a missing wedge of information”.

REVIEWER COMMENTS

Reviewer #1 (Remarks to the Author):

The authors provide more testing cases for the software. The results are convincing that the algorithm presented here is on par with the state-of-the-art methods. They also provide explanation why the earlier version of DeepDeWedge produces hallucinations or overpronounced details in the reconstructed tomograms.

Here are some minor issues:

The relative brightness of the panels in all Figures,3,4,5,7 in the revised paper is not the same, preventing the direct comparison. For example, in Figure 3 the DeepDewedge one is brighter than the tomogram processed with cryoCARE+IsoNet.

In figure 7, the cryoCARE + IsoNet result seems worse than the previous result only using IsoNet. Why that is the case? It would be good to retain the old IsoNet processed ones in Figure 4 and 7, either in main or in supplement.

Reviewer #1 (Remarks on code availability):

I tested the code from this paper. It has enough instructions for installing and running the application.

Reviewer #2 (Remarks to the Author):

The revised article has been further modified and added a comparison of results with respect to the previous version. In terms of overall performance, it is comparable to isonet or cryoCARE+isonet. From a purely missing wedge correction point of view, the test results presented by the authors are good and partially better than isonet, but from a denoise point of view, the performance is questionable and may suffer from excessive noise reduction. For the single-particle sample in Figure 3, the current noise reduction intensity is good because the blank is only noise or noise from the solvent. But for the cell slice samples, because the cells are filled with a variety of very small proteins, excessive noise reduction removes some of the small molecule features that are close to noise. For example, in Figures 4 and 5, the results of isonet's processing are softer and try to keep the original material features inside the cell and some original details in the background. In DeepDeWedge's results, while the

larger structural features are much clearer, some of the detail of the cellular contents is also lost.

The presentation of the results in the paper is more problematic and modifications are suggested. The purpose of this paper is MISSING WEDGE CORRECTION, denoise is relatively minor, then the main thing that should be compared is the xz and yz sections of the tomogram. However, in most of the comparison plots, the authors mainly compare the image quality in the xy plane, which cannot reflect the effect of missing wedge correction. Further, the regions of interest labeled by the authors in the xy plane are not visible in the xz and yz cross-sections of these plots, which makes the presentation of such images problematic. For example, in the first row of Figure 5, the authors focused on a vesicle within the xy cross section, but this vesicle is not visible in the corresponding zx and yz cross-sectional plots. vesicles in the zx and yz cross-sections are critical to the comparison of the results, in particular the effect of missing wedge correction. Comparison of vesicles seen in the xy plane is at best indicative of the effect of denoise, but largely irrelevant to missing wedge correction. Such a problem exists in most of the figures in this paper.

It is recommended that the xy, xz, and yz sections be labeled in each cross-section diagram with their respective positions in the other sections. figure 7 suffers from a very serious problem of this kind, with the relationships between the individual cross-section diagrams being confusing. For membrane structures, one might be more concerned with the ability to recover membranes parallel to the xy plane, which is an important indicator of the effectiveness of missing wedge correction.

With further refinement of the presentation and comparison of results, this article should be able to meet the requirements for publication.

Reviewer #2 (Remarks on code availability):

All results in the manuscript can be reproduced.

We also tested the program using our own results, some result were good and some were not. The performance is not as stable as isonet.

Responses to Reviewers' Comments from Round 2

Reviewer comments are in **black**, **our** responses are in **blue**

We thank the reviewers very much for testing our software and for their valuable comments, based on which we improved the presentation and discussion of our results.

Reviewer #1 (Remarks to the Author):

The authors provide more testing cases for the software. The results are convincing that the algorithm presented here is on par with the state-of-the-art methods. They also provide explanation why the earlier version of DeepDeWedge produces hallucinations or overpronounced details in the reconstructed tomograms.

Here are some minor issues:

The relative brightness of the panels in all Figures, 3, 4, 5, 7 in the revised paper is not the same, preventing the direct comparison. For example, in Figure 3 the DeepDeWedge one is brighter than the tomogram processed with cryoCARE+IsoNet.

We appreciate the reviewer's observation. The differences in brightness resulted from the outlier suppression strategy we used to improve image contrast. Initially, we clamped tomogram voxel values based on the tomogram-wide voxel standard deviation. In the second revision, we now clamped pixel values based on the 0.5% and 99.5% quantiles of each image. This adjustment resolved most brightness differences while maintaining consistent post-processing across images.

In figure 7, the cryoCARE + IsoNet result seems worse than the previous result only using IsoNet. Why that is the case?

When comparing the CryoCARE + IsoNet reconstructions shown in Figure 7 to the ones obtained using only IsoNet, we find that the CryoCARE + IsoNet reconstructions are less blurry and contain more high-frequency details, and thus look subjectively better. This is probably because the CryoCARE part of CryoCARE + IsoNet removes more noise than IsoNet alone which comes at the cost of losing some high frequency information. This visually better reconstruction is not reflected in the correlation-based metrics in this example, which might be since correlation based metrics can assign good values to images that are visually too smooth.

It would be good to retain the old IsoNet processed ones in Figure 4 and 7, either in main or in supplement.

In the second revision, we have included the original IsoNet results in Figure 7, and show the IsoNet reconstructions of the *C. reinhardtii* flagella in Appendix E.

Reviewer #1 (Remarks on code availability):

I tested the code from this paper. It has enough instructions for installing and running the application.

Reviewer #2 (Remarks to the Author):

The revised article has been further modified and added a comparison of results with respect to the previous version. In terms of overall performance, it is comparable to isonet or cryoCARE+isonet. From a purely missing wedge correction point of view, the test results presented by the authors are good and partially better than isonet, but from a denoise point of view, the performance is questionable and may suffer from excessive noise reduction. For the single-particle sample in Figure 3, the current noise reduction intensity is good because the blank is only noise or noise from the solvent. But for the cell slice samples, because the cells are filled with a variety of very small proteins, excessive noise reduction removes some of the small molecule features that are close to noise. For example, in Figures 4 and 5, the results of isonet's processing are softer and try to keep the original material features inside the cell and some original details in the background. In DeepDeWedge's results, while the larger structural features are much clearer, some of the detail of the cellular contents is also lost.

Balancing noise reduction while preserving high-frequency details, such as small molecules or small molecular features, is a common challenge in denoising, especially in high-noise regimes like cryo-ET. We agree that DeepDeWedge prioritizes stronger denoising, resulting in smoother reconstructions, whereas CryoCARE + IsoNet removes less noise, staying truer to the FBP reconstructions. We have added this observation to a new paragraph in the Discussion section in which we compare the denoising and missing wedge reconstruction performance of DeepDeWedge and CryoCARE + IsoNet.

The presentation of the results in the paper is more problematic and modifications are suggested. The purpose of this paper is MISSING WEDGE CORRECTION, denoise is relatively minor, then the main thing that should be compared is the xz and yz sections of the tomogram. However, in most of the comparison plots, the authors mainly compare the image quality in the xy plane, which cannot reflect the effect of missing wedge correction.

While we agree that missing wedge reconstruction is an important part of our work, we believe that denoising is equally important. The combination of both missing wedge reconstruction and denoising is the central aspect of our method.

Nevertheless, we thank the reviewer for pointing out that the missing wedge reconstruction aspect was somewhat underrepresented in the figures and discussion. To change this, we have modified all relevant figures such that the zoomed-in sub-plots now show parts from the x-z-slices, where missing wedge artifacts are most prominent in FBP reconstructions. In addition, each figure now also shows the central x-z-slice through the reconstructions' Fourier transforms, so that missing wedge reconstruction performance can be visually assessed in the Fourier domain.

Finally, we discuss missing wedge reconstruction in more detail in the aforementioned new paragraph in the Discussion section, where we compare the performance of DeepDeWedge with that of CryoCARE + IsoNet.

Further, the regions of interest labeled by the authors in the xy plane are not visible in the xz and yz cross-sections of these plots, which makes the presentation of such images problematic. For example, in the first row of Figure 5, the authors focused on a vesicle within the xy cross section, but this vesicle is not visible in the corresponding zx and yz cross-sectional plots. vesicles in the zx and yz cross-sections are critical to the comparison of the results, in particular the effect of missing wedge correction. Comparison of vesicles seen in the xy plane is at best indicative of the effect of denoise,

but largely irrelevant to missing wedge correction. Such a problem exists in most of the figures in this paper.

It is recommended that the xy, xz, and yz sections be labeled in each cross-section diagram with their respective positions in the other sections. figure 7 suffers from a very serious problem of this kind, with the relationships between the individual cross-section diagrams being confusing.

We thank the reviewer for making us aware of difficulties related to interpreting our figures. We have followed the reviewer's advice, and each slice in each figure in the new revision now contains a horizontal and a vertical line, indicating the positions of the other two slices. We have also redesigned the figures so that the objects we zoom in upon are visible in all slices. Both changes together should allow for a more holistic understanding of all reconstructed tomograms and the zoomed-in-upon objects.

For membrane structures, one might be more concerned with the ability to recover membranes parallel to the xy plane, which is an important indicator of the effectiveness of missing wedge correction.

We show how DeepDeWedge handles reconstructing an object that is almost perfectly parallel to the x-y-plane (orthogonal to the electron beam) in Appendix G. There, we found that DeepDeWedge produces a reasonable prediction, which is, however, unstable. This is expected because objects parallel to the x-y-plane are mostly masked out by the missing wedge, so their reconstructions are based on little data. Therefore, it is unclear to what extent the reconstructions of objects parallel to the x-y-plane can be trusted.

In cases where the object is only moderately or partially aligned with the x-y-plane, we found that DeepDeWedge produces good and stable reconstructions, see e.g., the zoomed-in region of Tomogram 3 in Figure 5 or the vesicles in Figure 7.

With further refinement of the presentation and comparison of results, this article should be able to meet the requirements for publication.

We thank the reviewer for expressing this positive opinion, and hope that our modifications have addressed all concerns regarding the presentation and discussion of our results.

Reviewer #2 (Remarks on code availability):

All results in the manuscript can be reproduced.

We also tested the program using our own results, some result were good and some were not. The performance is not as stable as isonet.

REVIEWERS' COMMENTS

Reviewer #1 (Remarks to the Author):

The authors revised the figure according to my suggestions, and added a discussion paragraph about the results from the deepDewedge and cryoCare+IsoNet. I now have done more thorough tests for deepDewedge not only using the tutorial dataset but also with my own dataset. In general, deepDeWedge can produce well denoised and missing-wedge corrected results for most tomograms.

I now concur with the author's discussion that "DeepDeWedge produces overall cleaner reconstructions CryoCARE + IsoNet results are more faithful to the FBP reconstructions, especially for high-frequency details." And I also agree with the other reviewer's observation that the many results from IsoNet have better details. For example, in Figure 4, the monomers of microtubule are better resolved in cryoCARE+IsoNet tomogram, while deepDewedge tomogram has better overall contrast. Although correlation-based metrics indicate that DeepDeWedge performs better, the "more faithful to the FBP reconstructions, especially for high-frequency details" aspect may be more important for biologists. As such, I would think the author's claim that the "deepDewedge sometimes better than cryoCARE and IsoNet" is a bit deceiving. It depends on whether the users want the tomograms to be cleaner or sharper.

Another observation from my test is that the deepDewedge can erase cryoEM densities. In Figures 4 and 5, small densities that might represent proteins are less prevalent in the DeepDeWedge-processed tomograms. When testing with my tomograms, I found that this effect is more severe (removing or blurring larger proteins) in low-defocus tomograms. I wonder if the authors have noticed this effect and can comment on it.

Reviewer #1 (Remarks on code availability):

I confirm that the results for tutorial dataset is reproducible, and the code is a usable and useful resource for the community.

Reviewer #2 (Remarks to the Author):

All my comments have been addressed. The manuscript is in a good shape to publish.

Responses to Reviewers' Comments from Round 3

Reviewer comments are in **black**, **our** responses are in **blue**

We thank the reviewers very much for their final comments on the latest revision of our paper and their constructive feedback throughout the entire review process, which have led to a significant improvement of the paper compared to the initial submission.

Reviewer #1 (Remarks to the Author):

The authors revised the figure according to my suggestions, and added a discussion paragraph about the results from the deepDewedge and cryoCare+IsoNet. I now have done more thorough tests for deepDewedge not only using the tutorial dataset but also with my own dataset. In general, deepDeWedge can produce well denoised and missing-wedge corrected results for most tomograms. I now concur with the author's discussion that "DeepDeWedge produces overall cleaner reconstructions CryoCARE + IsoNet results are more faithful to the FBP reconstructions, especially for high-frequency details." And I also agree with the other reviewer's observation that the many results from IsoNet have better details. For example, in Figure 4, the monomers of microtubule are better resolved in cryoCARE+IsoNet tomogram, while deepDewedge tomogram has better overall contrast. Although correlation-based metrics indicate that DeepDeWedge performs better, the "more faithful to the FBP reconstructions, especially for high-frequency details" aspect may be more important for biologists. As such, I would think the author's claim that the "deepDewedge sometimes better than cryoCARE and IsoNet" is a bit deceiving. It depends on whether the users want the tomograms to be cleaner or sharper.

We have removed the claim that "DeepDeWedge is sometimes better than CryoCARE+IsoNet" throughout the paper, and have replaced it with variations of "[DeepDeWedge] performs competitively and produces more denoised tomograms with higher overall contrast", as in the abstract, for example.

Another observation from my test is that the deepDewedge can erase cryoEM densities. In Figures 4 and 5, small densities that might represent proteins are less prevalent in the DeepDeWedge-processed tomograms. When testing with my tomograms, I found that this effect is more severe (removing or blurring larger proteins) in low-defocus tomograms. I wonder if the authors have noticed this effect and can comment on it.

We thank the reviewer for sharing this insight. The erasing effect may be a consequence of the strong denoising of DeepDeWedge. As denoising is often accompanied by a smoothing effect, small densities that are just above the noise level may become less visible.

This phenomenon is not unique to DeepDeWedge. In our experiments with IsoNet's built-in denoiser on EMPIAR-10045, we have made an observation that is similar to what is described by the reviewer: When the additional noise used in IsoNet's Noisier2Noise-like denoiser is high (which corresponds to stronger denoising), some ribosomes became more opaque or even partially vanished in the reconstruction.

Reviewer #1 (Remarks on code availability):

I confirm that the results for tutorial dataset is reproducible, and the code is a usable and useful resource for the community.

We thank the reviewer for testing our software and are pleased that the reviewer considers it useful.

Reviewer #2 (Remarks to the Author):

All my comments have been addressed. The manuscript is in a good shape to publish.

We thank the reviewer for the positive evaluation of our manuscript.